# Deep Learning Based Emboli Detection Using Ultrasound Doppler Imaging

**Raghava Vinaykanth Mushunuri**[*1,2]    RAGHAVA.V.MUSHUNURI@NTNU.NO
[1] *Department of Computer Science (IDI), Norwegian University of Science and Technology, Trondheim, Norway*

[2] *Department of Cardiology, St.Olavs, Trondheim, Norway*

**Cecilie Le Duc Dahl**[*1]    CECILIE.DAHL@LYSE.NET
**Elisabeth Krogstad Iversen**[*1]    ELIS.K.IVER@GMAIL.COM
**Sigrid Dannheim Vik** [3,4]    SIGRID.VIK@NTNU.NO
[3] *Department of Circulation and Medical Imaging (ISB), Faculty of Medicine and Health Sciences, NTNU*

**Martin Leth-Olsen**[3,4]    MARTIN.LETH-OLSEN@NTNU.NO
[4] *Children's Clinic, St Olav's University Hospital, Trondheim, Norway*
**Hans Torp**[3,5]    HANS.TORP@NTNU.NO
**Siri Ann Nyrnes**[3,5]    SIRI.A.NYRNES@NTNU.NO
[5] *Cimon Medical, Trondheim, Norway*

**Gabriel Kiss**[1]    GABRIEL.KISS@NTNU.NO

**Editors:** Accepted for publication at MIDL 2026

## Abstract

Accurate detection of embolic signals in the bloodstream is crucial for early diagnosis and prevention of cerebrovascular complications, and this work develops and evaluates an artificial intelligence–based system for automatic emboli detection in power Doppler imaging from NeoDoppler, aiming for robust and real-time performance. The study uses a four-stage experimental pipeline built on convolutional neural networks with transfer learning: an initial baseline model (Stage 1), an assessment of spatial generalisation (Stage 2), and a hybrid two-step strategy (Stage 3) that combines conventional High-Intensity Transient Signal (HITS) pre-detection with CNN-based classification, followed by a simplified preprocessing strategy in Stage 4, where single-channel images are replicated into three channels to match pre-trained CNN architectures; all models are trained with 5-fold cross-validation on 523 recordings from 25 patients and evaluated on unseen pilot recordings from the same cohort and additional abdominal surgery data. Across stages, performance improves progressively, with the hybrid two-step framework using the three-channel replication yielding strong results, achieving 96% sensitivity and 98% specificity on the pilot recording and 94% sensitivity and 71% specificity on the abdominal surgery recordings. We estimated 95% confidence intervals (CIs) using Wilson's score for abdominal surgery recordings, with a CI of 0.730-0.99, demonstrating that the proposed approach is an efficient and interpretable solution for ultrasound-based emboli monitoring.

**Keywords:** Emboli detection, NeoDoppler, Cardio Vascular Intervention, Explainable AI, High Intensity Transient Signals, Deep Learning, Transfer Learning

---

[*] Contributed equally

## 1. Introduction

Around 1% of all children are born with congenital heart defects. Of these, around 25% will require cardiac surgery or transcatheter intervention. During these interventions, there is a risk of emboli creation, which might enter the bloodstream (O'Brien et al., 1997; Rodriguez et al., 1998; Wallace et al., 2016; LaRovere et al., 2017). Although the clinical implications on children are unclear (Leth-Olsen et al., 2022), neurodevelopmental impairment in children with congenital heart disease is common (Marelli et al., 2016), and cerebral emboli can potentially be harmful. Therefore, neuromonitoring to prevent harmful hemodynamic events during interventions or surgery is important (Andropoulos et al., 2010). Hence, it is paramount to trace the emboli signals entering the bloodstream. Near-infrared spectroscopy has been employed for cerebral monitoring, but it cannot detect microemboli. Transcranial Doppler (TCD) is also used, but it is not helpful in the case of neonates as it requires larger and rigid head for the transducers (Durandy et al., 2011).

Cimon Medical has developed NeoDoppler (see Fig. 1), a new ultrasound device which is lightweight, small and can be gently attached to the fontanel for continuous blood flow monitoring. The Cimon device was clinically tested by Leth-Olsen et al. (2022) and Vik et al. (2020). NeoDoppler's potential in detecting high-intensity transient signals (HITS) containing emboli signals, artifacts, or both was demonstrated by visual inspection of the colour M-mode Doppler recordings. However, time-consuming manual analysis of recordings after intervention procedures is primarily intended for research purposes only (Cullinane et al., 2000). During the actual procedure, it is essential to detect air and solid emboli in real-time. However, this task is highly demanding for clinicians, as it requires them to continuously monitor for emboli while simultaneously performing surgical monitoring. Therefore, implementing automated detection systems becomes critical. These systems can provide real-time alerts to clinicians when emboli signals are detected and would allow for prompt responses and, as such, improve patient outcomes.

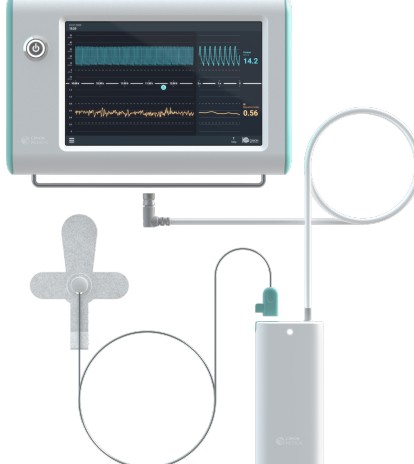

Figure 1: NeoDoppler by Cimon Medical

1

---

1. NeoDoppler device developed by Cimon Medical (https://cimonmedical.com/neodoppler/)

As mentioned above, the formation of emboli comes with its own risks during intervention procedures. Identifying emboli is crucial due to the inherent risks they pose during intervention procedures. Notable research has been done in the lines of conventional signal processing approaches (Cullinane et al., 2000; Geryes et al., 2016; Lam, 2022), but they have a problem in differentiating artifacts from air-emboli.

Cullinane et al. (2000) proposed a single-gate TCD emboli detection approach based on a fixed 3 dB intensity threshold above a median-filtered background signal, while Geryes et al. (2016) similarly applied a constant EBR threshold to spectrogram power for emboli identification. Although computationally simple, these fixed-threshold strategies are highly sensitive to noise and poorly adaptable to patient-specific signal variations, inevitably leading to a high rate of false detections (false positives). More recently, Lam (2022) applied a 15 dB EBR threshold to NeoDoppler power images for neonatal emboli monitoring; however, this method also struggles to reliably distinguish true emboli from artifacts, highlighting the inherent limitations of traditional threshold-based approaches.

Given these challenges, AI plays a crucial role in enhancing the detection of emboli. Convolution Neural Networks (CNNs), since its inception, has been expanding its application, and its footprint is now everywhere, ranging from space applications, underwater robotics, military, drones and many more. Although there is an upsurge in the applications of AI in the medical domain, research related to air-emboli detection is notably sparse (Sombune et al., 2017; Guépié et al., 2018; Grimstad, 2024). Recent advancements in CNNs have significantly improved medical image recognition, diagnosis, and treatment planning.

Sombune et al. (2017) classified embolic signals and artifacts using features from a fixed Doppler frequency range, reporting sensitivities and specificities of 83.0% and 80.1%, respectively. Guépié et al. (2018) explored traditional machine-learning methods such as SVM, Naive Bayes, and Decision Trees on TCD recordings, but performance was limited due to the susceptibility of TCD signals to motion and noise artifacts. More recently, Grimstad (2024) applied deep learning to NeoDoppler data, yet the model demonstrated poor generalisation to unseen pilot recordings, further highlighting the challenge of developing robust emboli detection systems.

In this project, we explored the potential of using AI to detect emboli in real-time within the blood flow during interventions, with a particular focus on detecting larger emboli, which are the ones believed to be of the most clinical significance, while also having a reasonable amount of false detections.

In this experimental study, we systematically examined the impact of three key factors on the performance of state-of-the-art (SOTA) deep learning models for emboli detection: 1) Spatial Position: assessing generalisation to off-center emboli; 2) Hybrid Architecture: evaluating a two-step framework integrating signal processing with deep learning; and 3) Input Processing: investigating the effect of simple three-channel grayscale replication. We trained and identified the best-performing model when compared to others using sensitivity and specificity as evaluation metrics. All the models are trained using 5 fold cross-validation and, were tested subsequently tested on a 30-minute unseen recording from the same cohort. The best performing model was finally tested on 6 recordings from 4 patients who underwent abdominal surgery. Notably, it successfully detected most of the emboli and effectively distinguished artifacts. We also evaluated the model's performance using explainable AI

methods (Saliency maps (Simonyan et al., 2013)) to determine whether the model made decisions based on the significant features of emboli signals.

## 2. Methods

### 2.1. Dataset

The dataset for this study includes Power Doppler images containing emboli, normal blood flow, and artifacts or noise signals. These images are extracted from a subset of recordings acquired by Leth-Olsen et al. (2022), who manually annotated all the recordings included in this experiment. Neonates with congenital heart diseases requiring transcatheter interventions or surgery with cardiac pulmonary bypass are continuously monitored using a customised ultrasound probe attached over the anterior fontanelle during the procedure, with prior consent from their parents. However, a few recordings were excluded due to the lack of annotations. All annotations are saved to a CSV file containing timestamps of emboli signal locations, the Emboli-to-Blood Ratio (EBR), and the background signal (BGS) intensity. A total of 523 recordings from 25 patients (12 from transcatheter interventions and 13 from surgery) are used in this project. Unseen recording from the same cohort and 6 additional recordings from 4 patients who underwent abdominal surgery (subset of the dataset from Vik et al. (2023) have also been used for further evaluation. All unseen recordings used for evaluation were approximately 30 minutes in duration.

From these recordings, we generated images containing air-emboli signals, which were classified as positive. In contrast, images depicting normal blood flow and artifacts are classified as negative images. Images of 1 second were generated from the timestamps provided in the annotations. The one-second window is generated with 25% overlap, and if the start and end time stamps of the windows do not contain annotations of the emboli signal, they are identified as the negative class, and if the annotation is present, it is identified as the positive class. To avoid class imbalance, negative samples were randomly subsampled to match the number of positive samples, based on the emboli annotations. The positive dataset was defined using an Emboli-to-Background Ratio (EBR) threshold $\geq 10.5$ dB and a Background Signal (BGS) threshold of $\geq 10$ dB. The negative dataset comprised two signal types, consisting of 80% artifacts and 20% blood flow images to simulate realistic noise conditions. These recordings also contain acquisitions at several depth levels. For this study, images were extracted from power Doppler signals at a depth of 12.8 mm, corresponding to the distance from the probe to the tissue. Fig. 2 portrays images corresponding to each class. For the experiments where emboli signals with offset are used, offset is added randomly to the window, so that the emboli signal is not centered (see Fig. 4).

We performed 5-fold cross-validation, ensuring that the dataset partitioning was done on the patient level for each cross-validation step so that there was no overlap between the training, validation, and test sets within each fold. A total of 1,383 images were used for the experiments, with 70% for the training set, 10% for the validation set, and 20% for the test set. All the experiments have $\geq 200$ testing samples. Unseen recordings were not included in the training phase and have been used only for evaluating the model's performance.

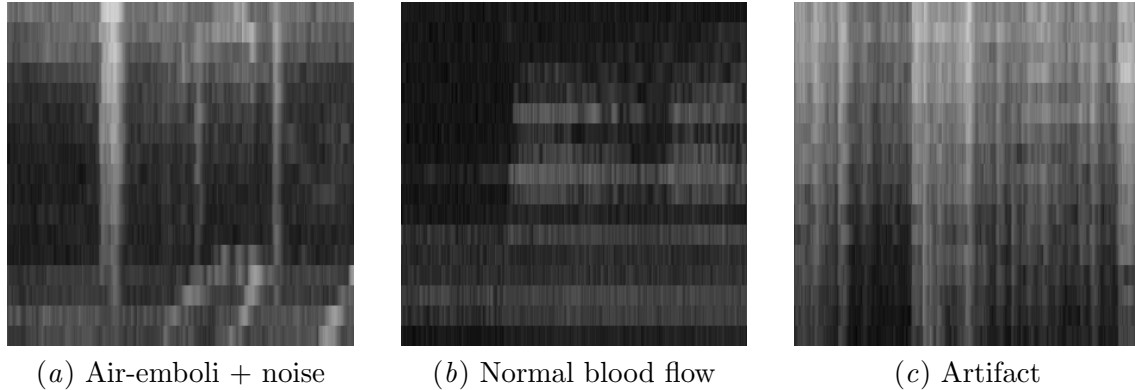

(*a*) Air-emboli + noise    (*b*) Normal blood flow    (*c*) Artifact

Figure 2: Example images for experiments: (a) shows both artifacts and embolic signals, (b) shows a blood flow signal, and (c) shows an artifact signal.

## 2.2. End-To-End pipeline

Our final pipeline, as portrayed in Fig.3, is a two-step process with a threshold-based algorithm to identify HITS and Non-HITS; the signals identified as HITS are then sent to the classification step. The best-performing model, evaluated on the pilot recording, is used for further evaluation on 6 additional unseen recordings. As described in the 2.1, these recordings are obtained from patients who underwent abdominal surgery. It contained 13,500 images, of which only 26 contained embolic signals. The remaining ones are either blood flow images or artifact images. Images are acquired with an offset and 25% overlap. The evaluation metrics are the same as mentioned in 2.4. To identify appropriate parameters for the dataset and the best-performing model, we conducted a four-step ablation study as described in the section below.

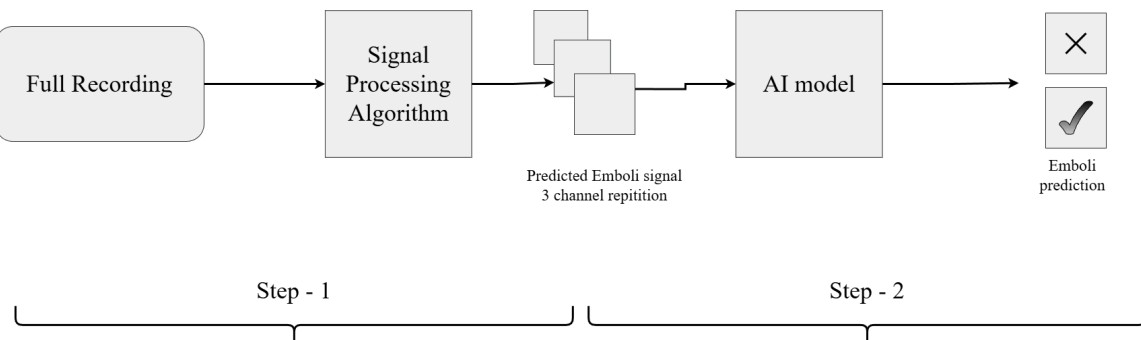

Figure 3: Visualisation of the two-step method.

In addition to using the model in an emboli alarm system, we also created a graphical user interface (GUI) that could support emboli research by assisting with the annotation of ultrasound recordings. This GUI loads the ultrasound recordings and applies Lam's algorithm in the first step, then passes the resulting data through our CNN model, and finally outputs the automatic detections generated by the system. To enable execution of

the model on the CPU, we exported the best-performing weights to an ONNX file and deployed it for inference and evaluation.

## 2.3. AI-based emboli detector: Ablation studies

Early and reliable detection of embolic signals in ultrasound recordings is crucial for preventing cerebrovascular complications. This section presents the development and evaluation of an artificial intelligence–driven emboli detection system that automates the identification of embolic events in power Doppler imaging. The proposed framework utilises convolutional neural networks (CNNs) with transfer learning to distinguish between embolic and non-embolic signals under various imaging and noise conditions.
The study is structured into four successive stages. The study is structured into four successive stages, each formulated as a controlled ablation experiment designed to evaluate how the model's behavior changes in response to a single, well-defined modification of the dataset.In Experiment 1, baseline CNN models are retrained using images where embolic events are temporally centerd, establishing the core detection performance. Experiment 2 assesses model generalisation by introducing emboli that appear at spatial or temporal offsets within the imaging frame. Building upon these results, Experiment 3 introduces a hybrid two-step detection framework that combines signal processing for initial event localisation with CNN-based classification to increase accuracy and reduce false positives. Finally, Experiment 4 investigates a simplified preprocessing approach in which the single-channel ultrasound images are directly replicated across three channels, enabling straightforward compatibility with pretrained CNN architectures.

Each stage incrementally improves model robustness, accuracy, and suitability for real-time clinical deployment in ultrasound-based emboli monitoring.

### 2.3.1. Centered Emboli

The first experiment evaluated the performance of various CNN architectures—VGG16 (Simonyan and Zisserman, 2014), ResNet101(He et al., 2016), EfficientNet-V2(Tan and Le, 2021), MobileNet-V3(Howard et al., 2019)—for detecting centered embolic signals in ultrasound power Doppler images.

Here, centered emboli refer to image segments extracted such that the embolic event is temporally centered within the sample window. Based on the manually annotated timestamps from the original ultrasound recordings, each embolic signal was extracted with a 0.5-second margin on either side of the event. This ensured that the embolus appeared approximately in the middle of each image and that contextual signal information before and after the embolus was preserved.

To adapt a single-channel model that expects RGB input, an additional convolutional layer was introduced to map the grayscale input to three channels. The pretrained feature extractor layers were kept frozen, and only the first and final classification layers were retrained. The classification head was replaced with a single-neuron output (without activation) for binary prediction. This experiment replicates Grimstad (2024) recent contributions to emboli detection using deep learning.

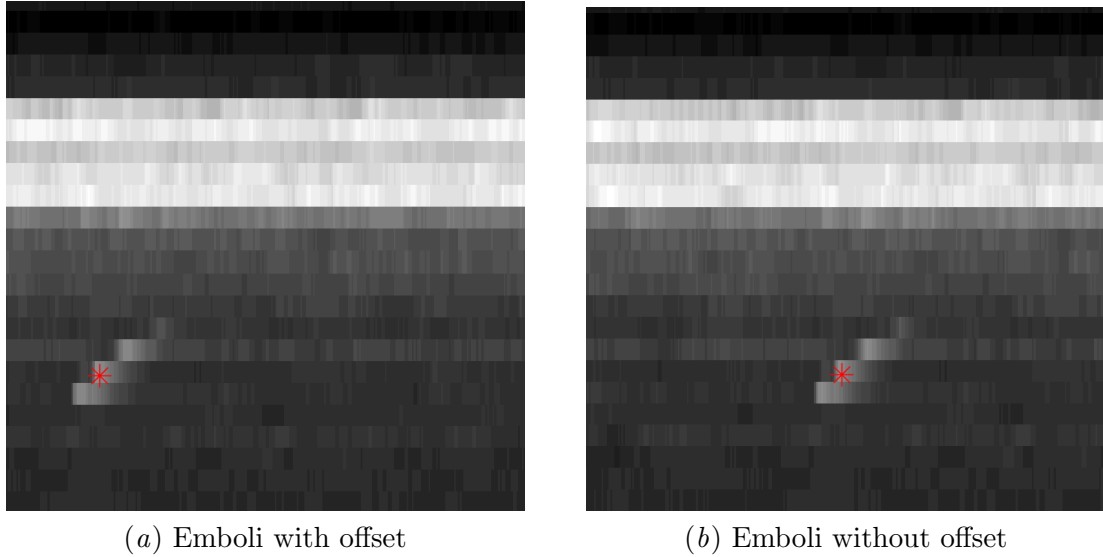

         ($a$) Emboli with offset                 ($b$) Emboli without offset

Figure 4: Example images for experiments: (a) shows emboli with Offset, (b) shows emboli without Offset.

### 2.3.2. **Emboli with offset**

The second experiment extended the baseline model to assess detection robustness when embolic events occurred at varying spatial or temporal offsets relative to the image center. Using the same architectures and training protocols as in Stage 1, the networks were retrained on a dataset containing off-center emboli while maintaining the same negative samples. This experiment evaluated spatial generalisation—specifically, the model's ability to maintain detection accuracy when emboli appear at different positions within the field of view. The difference between the emboli images with and without offset is shown in Fig. 4

### 2.3.3. **Two step approach**

After training models with and without emboli offset, we employed a two-step approach for next ablation experiments. The two-step detection framework consists of an initial signal-processing stage followed by deep-learning classification using grey scale spectrum. In the first step, high-intensity transient signals (HITS) are extracted from the raw Doppler recordings using the threshold-based method described by Lam (2022), generating candidate regions that may contain both embolic signals and noise. In the second step, the two best-performing CNN architectures from the earlier experiments are fine-tuned to classify each candidate region as embolic or non-embolic, using the same dataset configuration, cross-validation strategy, and training setup from section 2.3.2.

### 2.3.4. **Three channel replication**

The final experiment explored an alternative and computationally simpler preprocessing approach. Instead of applying a learnable convolutional layer to expand grayscale ultra-

sound images, the single channel was directly replicated across three channels to satisfy the input requirements of pretrained CNNs. This method preserved the intrinsic intensity distribution of the ultrasound data while simplifying the preprocessing pipeline.

Models were trained and validated on datasets encompassing emboli with offset and with a wide range of EBR values to assess the effect of signal intensity variation on classification performance. The dataset configuration is similar to that of 2.3.3, however the first trainable layer from earlier model configuration is now removed.

## 2.4. Evaluation

Evaluation is paramount to estimate the performance of the models and it is crucial in medical imaging domain, where misclassification can lead to risky repercussions. It is also crucial identify potential bias if present. There are two types of evaluation procedures namely, quantitative evaluation and qualitative evaluation. Quantitative evaluation is done using standard metrics, such as accuracy, sensitivity, specificity and F1 score. These metrics are calculated using a confusion matrix. Explainable AI (XAI) involves methods and techniques that help humans perceive the decision-making process of an AI model by providing insights on how it arrived at a particular decision/s (Samek et al., 2017; Gilpin et al., 2018). Saliency maps visualisation is a technique that highlights the regions of an input image that most strongly influence a model's prediction, offering a visual explanation of where the network focuses. Simonyan et al. (2013). In this project, we used this method to understand the model's behaviour when testing it on unseen recordings. We also estimated a 95% confidence interval (CI) using the Wilson score to understand how robust the model is in detecting true emboli events.

## 2.5. Training details

Networks are trained using the PyTorch framework on an NVIDIA RTX A6000 GPU, and the Torchvision library is used for pre-trained model weights from the Imagenet dataset. For XAI techniques, we have used the Captum library. Image dimensions are originally 289 $\times$ 291 $\times$ 1 and then reshaped to 256 $\times$ 256 $\times$ 1 using a nearest neighbour interpolation technique. Later, a convolution layer is added to obtain a three-channel (see 2.3.1) image of size 224 $\times$ 224 $\times$ 3, which is the standard image dimension expected by the models. Images are loaded using the PIL (pillow) library, and the images are divided by the maximum value to have a pixel range of 0-1.0. Batch size is set to 32, and a learning rate of $1 \times 10^{-2}$ is used. Binary cross-entropy loss is used as the loss function, and the ADAM optimiser is used for optimising model weights. Models are initially trained for 400 epochs, but later we changed to 200 as models are converging around 200 epochs. All the models are trained for 5-fold cross-validation, and the dataset is divided accordingly on the patient level.

## 3. Results and Evaluation

### 3.1. Quantitative evaluation of experiments from ablation study

An ablation study is designed to examine the model's behaviour under various dataset parameter changes. In this ablation study, each experiment is independent of the others. All the models from each ablation experiment are evaluated quantitatively using standard

evaluation metrics specified in sec. 2.4, and qualitative analysis is done on unseen recordings using XAI methods as specified in sec. 2.4. The results of each ablation experiment are described below.

### 3.2. Ablation-Experiment-1: Images with centered emboli signals

All SOTA models were evaluated using 5-fold cross-validation, and the best-performing model from the 5 folds was evaluated on a Pilot recording, which is not part of the training and validation phase. All the models were evaluated using evaluation metrics as described in 2.4 and are summarised in Tab. 1. VGG16 and ResNet101 models performed well during the validation phase; however, when tested on pilot recordings, the models failed to identify various signals because the emboli signals are not centerd in unseen recordings. Hence, in the next step we have trained the models with offset emboli signals.

| Ablation-Experiment 1 | | | | | | | | |
|---|---|---|---|---|---|---|---|---|
| Model | Accuracy | Precision | Sensitivity | Specificity | TP | FP | FN | TN |
| VGG16 | 65% | 3% | 72% | 64% | 26 | 840 | 10 | 1522 |
| EfficientNet | 95% | 11% | 33% | 96% | 12 | 94 | 24 | 2268 |
| ResNet101 | 96% | 19% | 50% | 96% | 18 | 75 | 18 | 2287 |
| MobileNet | 95% | 7% | 19% | 96% | 7 | 93 | 29 | 2269 |
| DenseNet | 82% | 4% | 50% | 83% | 18 | 404 | 18 | 1958 |

Table 1: Evaluation of first ablation experiment with centered emboli signals.

We further evaluated the performance of the approach from experiment-4 qualitatively using the saliency maps method with pilot-recording, and the heatmaps are shown in Figure. 5.

### 3.3. Ablation-Experiment-2: Emboli signals with offset

To observe the change in model performance, we trained all models from Experiment 1 using offset emboli signals as described in section 2.3.2. The models performed with nearly perfect accuracy during validation. Although the evaluation showed improved performance compared to the previous experiment stage, they still failed to reduce false positives when evaluated using the pilot recording. The evaluation metrics are summarised below in Tab. 2

| Ablation-Experiment 2 | | | | | | | | |
|---|---|---|---|---|---|---|---|---|
| Model | Accuracy | Precision | Sensitivity | Specificity | TP | FP | FN | TN |
| VGG16 | 11% | 2% | 94% | 10% | 34 | 2131 | 2 | 231 |
| EfficientNet | 81% | 1% | 17% | 82% | 6 | 433 | 30 | 1929 |
| ResNet101 | 19% | 2% | 83% | 18% | 30 | 1934 | 6 | 428 |
| MobileNet | 52% | 2% | 58% | 52% | 21 | 1141 | 15 | 1221 |
| DenseNet | 17% | 2% | 92 % | 16% | 33 | 1984 | 3 | 378 |

Table 2: Evaluation of second ablation experiment with offset emboli signals.

### 3.4. Ablation-Experiment-3: Two-step approach

To further reduce the false positives, we used two steps. In the first step, we filtered the pure blood flow signals from HITS, and in the second step, we classified the HITS as emboli or artifact signals using deep learning algorithms. For this, we have chosen the VGG16 and ResNet101 models for training, as they both performed well in the previous experiment stages. The evaluation metrics are shown in Tab. 3

| Ablation-Experiment 3 | | | | | | | | |
|---|---|---|---|---|---|---|---|---|
| **Model** | **Accuracy** | **Precision** | **Sensitivity** | **Specificity** | **TP** | **FP** | **FN** | **TN** |
| VGG16 | 96% | 51% | 96% | 96% | 25 | 24 | 1 | 561 |
| ResNet101 | 94% | 40% | 88% | 94% | 23 | 35 | 3 | 550 |

Table 3: Two-step method on the Pilot recording

### 3.5. Ablation-Experiment-4: Three channel replication

| Ablation-Experiment 4 | | | | | | | | | |
|---|---|---|---|---|---|---|---|---|---|
| **Model** | **Parameter** | **Accuracy** | **Precision** | **Sensitivity** | **Specificity** | **TP** | **FP** | **FN** | **TN** |
| VGG16 | EBR 10.5 | 98% | 68% | 96% | 98% | 25 | 12 | 1 | 573 |
| VGG16 | EBR 7 | 95% | 45% | 96% | 95% | 25 | 31 | 1 | 554 |
| VGG16 | EBR 4 | 98% | 75% | 92% | 99% | 24 | 8 | 2 | 577 |
| ResNet101 | EBR 10.5 | 98% | 68% | 96% | 98% | 25 | 12 | 1 | 573 |
| ResNet101 | EBR 7 | 97% | 60% | 96% | 97% | 25 | 17 | 1 | 568 |
| ResNet101 | EBR 4 | 97% | 57% | 92% | 97% | 24 | 18 | 2 | 567 |

Table 4: Evaluation metrics of VGG16 and ResNet101 after swapping to three-channel replication (experiment-4).

Although the two-step approach performed well, there is still room for improvement. We further experimented by replicating the grey-scale array into 3 channels and observed a significant increase in specificity. We focused solely on improving specificity, thereby reducing false positives. As shown in Tab. 4, unseen recording testing shows a greater reduction in false-positive predictions.

### 3.6. Evaluation of end-end pipeline

From the above ablation studies, it is observed that the two-step approach and 3-channel replication significantly improved the detection of emboli signals when tested on the unseen pilot recording. In particular, ResNet101 performs better at detecting emboli across various EBR thresholds. We have used the ResNet101 model for testing on the abdominal surgery dataset.

The two-step pipeline with the ResNet101 model trained using three-channel replication is evaluated on unseen recordings from abdominal surgery that were not part of the training and validation cycles, using the same metrics. Here, the sensitivity and specificity are

estimated on the model output, that is, we excluded the detections from the first step. The Wilson score is used to estimate a 95% CI for sensitivity and is reported in Table 5.

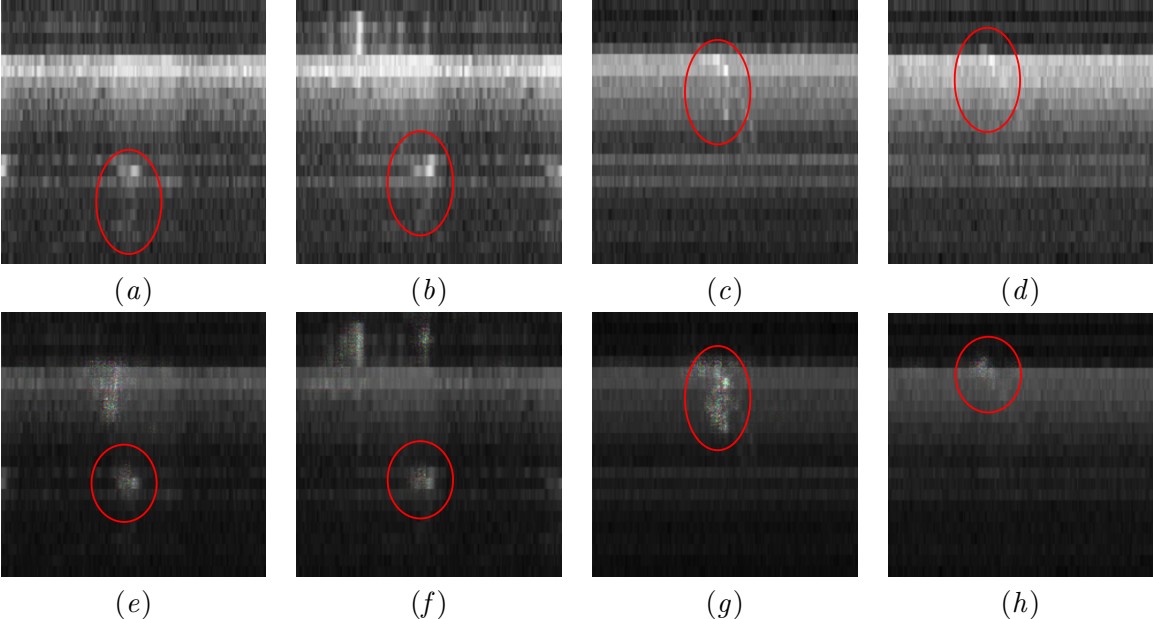

Figure 5: XAI heatmaps for true positive predictions with unseen recording: subfigures (a), (b), (c), and (d) are the original images, while (e), (f), (g), and (h) are the corresponding heatmaps overlaid on the original images. Emboli regions and saliency maps are highlighted in both the original and the overlaid maps.

When tested on the pilot recording from the same cohort, the two-step approach only missed one emboli signal, as shown in Tab. 4. When this approach is tested on recordings from abdominal surgery, we missed 7 emboli signals in the first stage. We manually added the missed emboli signals to test whether the model without a filtering algorithm can detect them. The model detected all 7 missed signals. Evaluation metrics on unseen recordings are shown in Tab. 5. We later designed an alarm system to estimate the false-positive rate per minute during this evaluation phase and observed that our approach produced only 7% of the total false alarms compared (see Tab. 6) to the baseline model from experiment-1 by Grimstad (2024).

## 4. Discussion

The aim of this work is to detect emboli entering the cerebral circulation in real time using deep learning. We conducted an ablation study by training state-of-the-art (SOTA) models at various stages and observing how their performance changed with different dataset modifications. Across all experiments, the results consistently highlight both the promise and the challenges of applying CNNs to emboli detection. Experiment 1, which replicates Grimstad (2024) work, demonstrated that deep learning models can be utilised to detect emboli signals. When the models are tested using 5-fold cross-validation, the metrics were observed to be near perfect values, however, when tested on pilot unseen recording, the

| | Accuracy | Precision | Sensitivity | Specificity | TP | FP | FN | TN | Total negative | 95% CI of Sensitivity (Wilson Score) |
|---|---|---|---|---|---|---|---|---|---|---|
| Without intervention | 82% | 73% | 94% | 71% | 16 | 6 | 1 | 15 | 13540 | 0.73-0.99 |
| With intervention | 85% | 81% | 96% | 71% | 25 | 6 | 1 | 15 | 13549 | 0.81-0.99 |

Table 5: Evaluation metrics of ResNet101 on new test data.

| | False alarms | False alarms/minute |
|---|---|---|
| Baseline | 88 | 2.9 |
| Our work | 7 | 0.23 |

Table 6: A comparison of the false alarm rate between Grimstad's model and our model.

models raised a massive number of false alarms and we also observed that the main drawback of this experiment is that it is biased towards the position of emboli signals, as the models were trained on centerd emboli signals. To overcome the drawbacks of this experiment, we trained the models with offset emboli and achieved better performance than those trained in Experiment 1. Experiment 2 demonstrated that positional robustness can be achieved, ensuring models rely on embolic features rather than fixed spatial features. We observed that sensitivity and specificity improved gradually when tested on a pilot recording. However, all the models still generated a high amount of false positives. For further experiments, we selected VGG16 and ResNet101 models, as there is no substantial improvement in the performance of other models. Experiment 3 further improved performance by integrating a two-step approach, in which classical signal processing substantially reduced false positives without compromising sensitivity, underscoring the need for additional improvements to achieve a robust model. Pertaining to the above reasons, we introduced a two-stage classifier model, as explained in Section 3 of the methods. This method reduced false positives further and improved sensitivity in a pilot recording, but the false-positive rate remains considerable. Experiment 4 demonstrated that simple three-channel replication is both more stable and effective than convolutional projection, significantly reducing false positives and proving to be a practical input strategy. ResNet101 model performed well during the evaluation phase. Additionally, we varied EBR thresholds to assess the model's performance across different emboli signal strengths. For that we chose EBR values as 10.5, 7 and 4 dB. Leth-Olsen et al. (2022) classified the signals with EBR≥15 dB as large emboli, and Lam (2022) employed this threshold in her algorithm, and Kjelsaas (2020) in their approach employed EBR≥9 dB; hence, we chose signals above 10.5 dB to be larger signals. When EBR is reduced, it includes smaller signals; hence, we choose 4 and 7 dB as two additional threshold values. EBR≥4 dB implies that the dataset contains all signals; EBR≥7 dB excludes small emboli signals. Emboli signals at different thresholds are shown in the picture below. XAI evaluation using saliency maps on the unseen recording also showed

that the model is looking at the patterns in the emboli signals and making predictions, indicating that the models are not predicting garbage and that the projections are based on meaningful features (see Figure. 5).

While evaluating the end-to-end pipeline, we leveraged the advantage of a two-step approach with a conventional thresholding algorithm for filtering HITS in the first step and 3-channel replication in the second step. We evaluated this two-step approach on 6 unseen recordings with a signal and designed an alarm system to track false-alarm frequency per minute, finding it as low as 0.23 per minute. We estimated the confidence interval of the sensitivity in identifying emboli signals using the Wilson score. The estimated 95% CI is reported in the Tab.5 for both the approaches. Although the emboli event rate is low across the six test recordings, the 95% CI ranges from 0.73 to 0.99 (without intervention) and 0.81 to 0.99 (with intervention), indicating fairly robust performance. However, this method requires a larger dataset to further validate the model's performance.

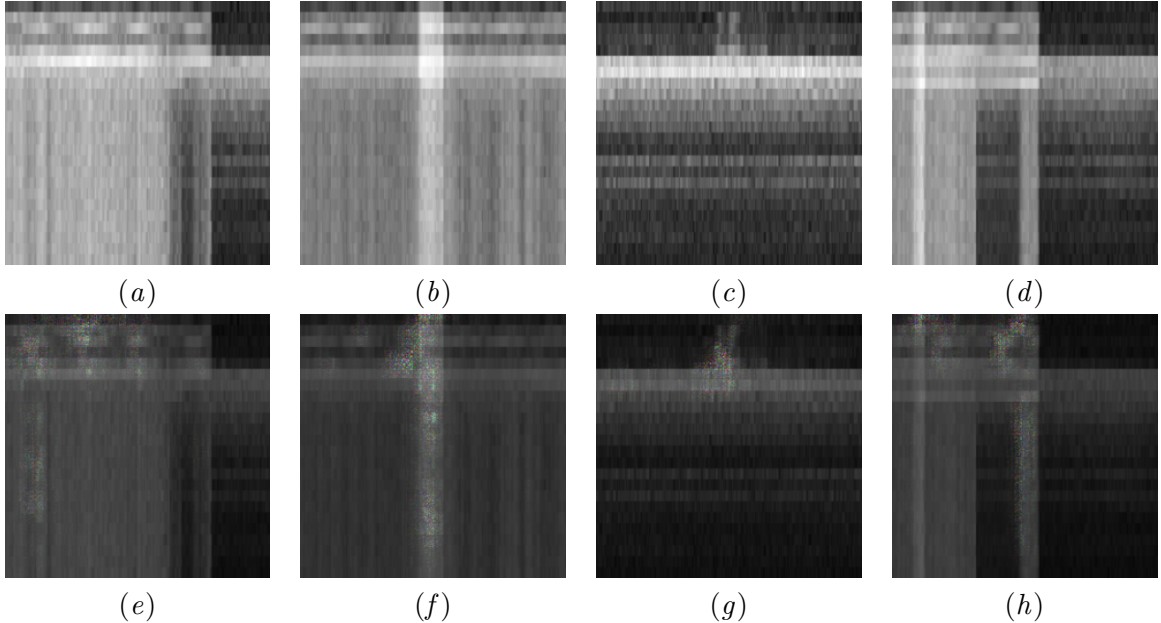

Figure 6: XAI heatmaps for False positive predictions with unseen recording: subfigures (a), (b), (c), and (d) are the original images, while (e), (f), (g), and (h) are the corresponding heatmaps overlaid on the original images. Emboli regions and the saliency maps are highlighted in both original and overlayed maps.

Although the two-step approach performed well during initial evaluation, when tested on unseen recordings from abdominal surgery, it missed a few emboli signals. This could be because the emboli signals in those recordings have lower EBR thresholds, and Lam's algorithm works better for higher EBR values. Our model also misinterpreted a few artifacts as emboli signals. XAI images of false-positives are shown in the figure 6. Although the false-positive rate is very low, the model identified a few noise patterns as emboli signals because they appear tilted, a significant feature of an emboli signal. Additionally, further study on the implications of artefacts due to motions as well as poor signal quality due

to the probe placement is not completely studied as the dataset contains such. We also designed a MATLAB-based GUI to load the recording and detect the emboli signals. For this, the models are converted to ONNX format, enabling them to run on CPUs and GPUs. The testing was conducted seamlessly with no performance degradation.

## 5. Conclusion

In conclusion, we developed a two-step approach to detect emboli signals in the bloodstream. The first step is a simple signal processing algorithm which detects HITS, and the second step identifies whether the filtered signal is an emboli or an artifact. We also integrated an ONNX model for inference using a CPU with a simple GUI.

## Acknowledgments

The authors would like to acknowledge the contributions of Arild Grimstad who implemented the initial experiments during his master thesis. Raghava Vinaykanth Mushunuri has received funding from the Liaison Committee for Education, Research and Innovation in Central Norway (project number:2023-34188).

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
