# OpenReview forum: "Deep Learning Based Emboli Detection Using Ultrasound Doppler Imaging"
_MIDL.io/2026/Validation_Papers — MIDL 2026 - Validation Papers Poster_

### Official Review · Reviewer_xfiZ · 2025-12-20

**Confidence:** 4
**Preliminary Rating:** 5

**Summary:**

This paper presents a deep learning-based system for detecting embolic signals in neonatal and adult ultrasound Doppler recordings. The authors design a four-stage pipeline using CNNs with transfer learning, including baseline models, spatial generalisation testing, a hybrid two-step approach combining signal processing with CNN classification, and three-channel grayscale replication for pretrained networks. Models are trained on 523 recordings from 25 patients and evaluated on unseen pilot recordings and abdominal surgery data. The proposed approach achieves high sensitivity (up to 96%) and specificity (up to 98%) on pilot data, while substantially reducing false positives compared to previous methods. This work demonstrates a practical, interpretable solution for real-time emboli monitoring with potential clinical impact.

**Strengths:**

Clinical Relevance: The study addresses an important medical problem early emboli detection in neonates and surgical patients to improve patient outcomes.

Methodological Rigor: The four-stage pipeline is systematically evaluated with cross-validation and unseen datasets, providing robust evidence of model performance.

Innovation: Combining classical HITS detection with CNN classification in a hybrid approach and introducing three-channel replication for single-channel ultrasound images shows thoughtful innovation to improve accuracy and reduce false positives.

Interpretability: Use of saliency maps and explainable AI techniques allows verification that the model focuses on clinically relevant features, increasing trustworthiness.

Practical Deployment: Exporting the model to ONNX for CPU inference and integrating a GUI demonstrates translational potential for clinical use.

**Weaknesses:**

Limited Dataset Diversity: While the dataset includes 523 recordings, the number of patients is relatively small (25 patients), potentially limiting generalization across populations and clinical settings.

Low Specificity in Certain Scenarios: Despite improvements, the model still shows reduced specificity in abdominal surgery recordings, especially for low EBR emboli (<4 dB), highlighting a limitation in detecting subtle events.

Comparative Evaluation: Although the method is compared to prior work (Grimstad, 2024), broader benchmarking against additional state-of-the-art emboli detection algorithms would strengthen claims of superiority.

Detail on Parameter Selection: Some hyperparameter choices, such as EBR thresholds and convolutional replication, could benefit from more justification or sensitivity analysis.

Data Imbalance Considerations: While the authors mention 80% artefacts in the negative class, more explicit discussion of class imbalance handling could clarify potential bias in training.

**Detailed Comments:**

Figures could include more examples of false positives and false negatives to illustrate model limitations.

Consider discussing potential impact of motion artefacts or probe placement variability on model robustness.

Minor language improvements would increase readability; some sections are dense with repetition.

Expanding discussion on deployment in real-time clinical workflows (e.g., latency, integration with monitors) would enhance translational value.

**Justification Of The Preliminary Rating:**

This paper presents a significant contribution to automated emboli detection using deep learning. Its novelty lies in the hybrid two-step detection method, systematic evaluation across multiple stages, and practical deployment with interpretable outputs. Despite dataset size limitations and some reduced specificity in challenging scenarios, the work shows strong scientific merit, clinical relevance, and translational potential. It is well-structured, methodologically rigorous, and likely to stimulate further research in medical AI for real-time ultrasound analysis.

**Questions To Address In The Rebuttal:**

How does the model generalize to a larger and more diverse patient population beyond 25 patients?

Can the authors provide more details on handling low EBR emboli and improving specificity in such cases?

Are there plans to compare this method against additional SOTA emboli detection approaches beyond Grimstad (2024)?

---

### Official Review · Reviewer_L3jD · 2025-12-25

**Confidence:** 3
**Preliminary Rating:** 3
**Final Rating:** 3

**Summary:**

The paper presents a deep learning-based system for detecting air emboli in NeoDoppler ultrasound recordings, aiming to prevent cerebrovascular complications in neonates undergoing cardiac interventions. By converting 1D Doppler signals into 2D Power Doppler images, the authors leverage image-based deep learning architectures to address this challenge. The authors propose a four-stage experimental pipeline, evolving from baseline CNNs to a hybrid approach that combines signal processing (HITS detection) with a classifier (ResNet101/VGG16) using a three-channel replication preprocessing strategy. The method is validated on 523 recordings from 25 patients and further tested on unseen pilot recordings and an out-of-distribution dataset.

**Strengths:**

1. They demonstrate a logical, step-by-step progression that justifies their design choices.
2. Have the experinment in out of distribution data.
3. The authors also focused on real-world clinical performance and tested the speed at which the model could be deployed in clinical practice.

**Weaknesses:**

1. In Experiment 2, the authors retrain models with "emboli with offset" to assess spatial generalization. While this improves performance, it is unclear how this conceptually differs from standard data augmentation techniques (e.g., random translation or cropping) commonly used in deep learning training pipelines.
2. The manuscript mentions ensuring "no class imbalance" for all experiments  and using 5-fold cross-validation. However, the external validation reveals an extreme real-world imbalance. The authors should clarify how the training data was balanced.
3. While the inclusion of the abdominal surgery dataset is a strength, the statistical power of this evaluation is limited. The text states that out of 13,500 images in this test set, "only 26 were images that contained embolic signals". Drawing strong conclusions about sensitivity (96% vs 94%) based on such a small number of positive events (missing just 1 or 2 emboli drastically changes the metrics) is risky.
4.The model's baseline is relatively weak and does not incorporate the recent models.

**Detailed Comments:**

The "Short Title" in the running header of the PDF is currently empty. There are several punctuation errors throughout the manuscript.

**Justification Of Final Rating:**

I would like to thank the authors for their detailed responses and the revisions made to the manuscript during the rebuttal period.
Resolved Issues: The authors have adequately addressed my concerns regarding:
1. Baseline Models & Data Augmentation: I accept the justification that lightweight models were chosen due to hardware constraints (CPU deployment) and that standard data augmentation was avoided to preserve the physical realism of blood flow signals.
2. Data Partitioning: The clarifications added to the methods section have improved the reproducibility of the study.
Remaining Concerns & Reason for Score: However, my primary concern regarding the statistical robustness of the external validation remains. While I appreciate the authors' effort to calculate the Confidence Intervals (CIs) as requested, the results (Sensitivity CI: 0.73–0.99) effectively confirm my initial apprehension. A confidence interval spanning from 73% to 99% indicates a high degree of uncertainty. This wide range is a direct result of the extremely small number of positive events (only 26 images) in the test set.
Conclusion: Although I recognize the significant challenges in collecting pediatric data, from the perspective of a Validation Study, the current statistical evidence is insufficient to robustly support the claims of high generalization performance. The risk that the reported high sensitivity is an artifact of the small sample size remains too high to warrant a higher score at this stage.
Therefore, while acknowledging the clinical value and the logical pipeline of the work, I maintain my rating of 3 (Borderline).

**Justification Of The Preliminary Rating:**

The paper aligns well with the scope of the Validation Studies Track, offering a pragmatic, clinically oriented solution for emboli detection. I appreciate the systematic four-stage experimental design and the effort to validate the model on an out-of-distribution dataset (abdominal surgery), which demonstrates a strong commitment to testing robustness. However, I have assigned a borderline rating due to several concerns regarding statistical robustness and reporting clarity: The external validation, while a strong concept, relies on an extremely small number of positive samples; It is unclear how "Experiment 2" (offset emboli) differs conceptually from standard data augmentation, and whether the data splitting strategy strictly maintained patient-level independence; The baseline method is relatively weak.

**Questions To Address In The Rebuttal:**

1. Regarding the external validation on abdominal surgery data, Can you provide Confidence Intervals (e.g., 95% CI) for the reported Sensitivity and Specificity? Given that missing just 1-2 emboli would significantly alter the sensitivity metric, how confident are you that this sample size is sufficient to demonstrate generalization?
2. Add more baseline methods, and report the data augmentation method's performance;
3. A clearer explanation of data partitioning.

---

### Official Review · Reviewer_g8pC · 2025-12-31

**Confidence:** 3
**Preliminary Rating:** 3
**Final Rating:** 4

**Summary:**

This paper studies automatic emboli detection in ultrasound Doppler imaging using deep learning, motivated by the need for real-time monitoring during neonatal and surgical procedures. The authors propose a four-stage experimental pipeline that explores offset training, a hybrid two-step detection framework combining HITS-based candidate selection with CNN classification, and a simplified three-channel replication strategy for transfer learning. Experiments are conducted on annotated NeoDoppler recordings from neonatal cardiac procedures and a small set of abdominal surgery data. The results suggest potential improvements in sensitivity and false alarm rates.

**Strengths:**

1. The paper addresses a clinically important and relatively underexplored problem: automated emboli detection in neonatal ultrasound Doppler imaging.
2. The use of real clinical data, including unseen recordings and a different surgical context, increases the practical relevance of the study.
3. The idea of combining classical signal processing (HITS detection) with deep learning classification is reasonable and aligns with clinical constraints on false alarms.
4. The paper is generally well structured, clearly written, and provides an adequate overview of prior peer-reviewed work in emboli detection.

**Weaknesses:**

1. The experimental design makes it difficult to attribute performance gains to individual components. Across the stages described in Section 2.2, multiple factors change simultaneously (candidate selection, preprocessing, and task definition), limiting the interpretability of comparisons in Tables 1 and 2.
2. Key evaluation metrics are not clearly defined. In particular, the precision values reported in Table 1 (1–3%) are difficult to interpret without a clear statement of whether metrics are computed at the image, event, or alarm level, and how overlapping windows are handled.
3. The paper lacks systematic ablation studies. Claims regarding the benefits of the two-step framework (Section 2.2.3) and three-channel replication (Section 2.2.4) are not supported by controlled experiments isolating each factor.
4. Evaluation on highly imbalanced data (Section 3.5, Table 3) relies mainly on accuracy and specificity, which are insufficient on their own to support strong conclusions in this setting.

**Detailed Comments:**

1. Adding a dedicated ablation section would significantly improve the paper.
2. Event-level or alarm-level metrics should be reported consistently across all evaluations.
3. Clarify how EBR thresholds were selected and whether they were tuned on validation data.

**Justification Of Final Rating:**

The authors have addressed the concerns raised in the initial review by providing clearer explanations of their experimental design and evaluation metrics. The inclusion of ablation studies and clarification on how metrics were computed enhances the interpretability of the results. The study remains relevant and contributes to the field of automated emboli detection in ultrasound Doppler imaging. While some limitations remain, the revisions have strengthened the manuscript sufficiently to warrant acceptance.

**Justification Of The Preliminary Rating:**

The paper addresses an important clinical problem and presents a method that appears promising, particularly in reducing false alarms for emboli detection. However, the current experimental design and evaluation methodology limit the strength of the conclusions. The lack of controlled ablations and unclear metric definitions make it difficult to assess which components are truly responsible for the reported improvements. These issues are potentially fixable with additional experiments and clarifications.

**Questions To Address In The Rebuttal:**

1. Can the authors provide ablation experiments that isolate the contribution of offset training, the two-step framework, and three-channel replication?
2. How exactly are precision, sensitivity, and specificity defined and computed, particularly in the presence of overlapping windows?
3. How robust are the reported results to changes in EBR thresholds and extreme class imbalance?

---

### Author Rebuttal · Authors · 2026-01-24

**Rebuttal:**

Respected chair and reviewers,

We would like to appreciate the time and efforts in reviewing our paper. With great diligence, we read the reviews and we tried to carefull address the comments and rebuttal questions by answering them using the official comments section and also made necessary changes in the manuscript and have also submitted the supporting material here. We have improvised Dataset section to clearly explain how the dataset is generated partitioned and fed for model training. In our earlier version of manuscript it was not quite clear about the ablation experiments we conducted, hence we changed the methods section as well as results section, to elucidate the ablatoin study. We also divided the table and added table corresponding to each experiment and clearly explainedd how we chose the next step to overcome the short comings of the experiments at each step. We also added explainable AI images for false positives to show the potential pitfalls of the model's performance. We also discussed potential issues such as the degradation of model's performance due to probe placement variability and poor quality signals in the discussion section.  We addressed the issues of readability by correcting the grammatical mistakes and also improved the readability by removing the repititions as much as possible. The modified sections of the manuscript are highlighted in yellow in the latest document.

We tried to improve upon the previous submission with the valuable reviews we recieved from the reviewers and we are hopeful that our paper adds value to the scientific community working with Ultrasound applications and AI. We are looking forward for further reviews to improve the manuscript even further.

**Supporting Material:**

/attachment/7d74d426adfefec8c0d91800a455a30445479e60.pdf

---

### Meta-Review · Area_Chair_DBNx · 2026-02-08

**Recommendation:** Accept (Oral)
**Confidence:** 5

**Metareview:**

The paper addresses an important clinical problem and presents a well-motivated hybrid approach for emboli detection in ultrasound Doppler imaging. Reviewers highlighted the relevance of the application, the use of real clinical data, and the systematic experimental pipeline as key strengths.

The reviewers raised concerns regarding the clarity of the experimental design, metric definitions, and the statistical robustness of the external validation due to the small number of positive events. The authors addressed these points in the rebuttal by clarifying data partitioning and evaluation metrics, explicitly restructuring the experiments as ablation studies, and adding confidence interval analysis for sensitivity and specificity. While limitations related to dataset size remain, they are now clearly acknowledged and discussed.

Overall, the rebuttal adequately addresses the major concerns, and the remaining issues do not outweigh the contribution and practical relevance of the work.

---

### Decision · Program_Chairs · 2026-02-14

Accept (Poster)